# Hierarchical Graph-Convolutional Variational Autoencoding for Generative Modelling of Human Motion

## Abstract

Models of human motion focus either on trajectory prediction or action classification but rarely both. The marked heterogeneity and intricate compositionality of human motion render each task vulnerable to the data degradation and distributional shift common to real-world scenarios. A sufficiently expressive generative model of action could in theory enable data conditioning and distributional resilience within a unified framework applicable to both tasks as well as facilitate data synthesis. We propose a novel architecture for generating a holistic model of action based on hierarchical variational autoencoders and deep graph convolutional neural networks. We show this Hierarchical Graph-convolutional Variational AutoEncoder (HG-VAE) to be capable of detecting out-of-distribution data, and imputing missing data by gradient ascent on the model's posterior, facilitating better downstream discriminative learning. We show that scaling to greater stochastic depth generates better likelihoods independently of model capacity. We further show that the efficient hierarchical dependencies HG-VAE learns enable the generation of coherent conditioned actions and robust definition of class domains at the top level of abstraction. We trained and evaluated on H3.6M and the largest collection of open source human motion data, AMASS.

## 1 Introduction

Human motion is naturally intelligible as a time-varying graph of connected joints constrained by locomotor anatomy and physiology. An understanding of human motion is necessary for tasks such as pose estimation, action recognition, motion synthesis, and motion prediction, across a wide variety of applications within healthcare Geertsema et al. (2018); Kakar et al. (2005), physical rehabilitation and training Chang et al. (2012); Webster & Celik (2014), robotics Koppula & Saxena (2013b;a); Gui et al. (2018b), navigation Paden et al. (2016); Alahi et al. (2016); Bhattacharyya et al. (2018); Wang et al. (2019), manufacture Švec et al. (2014), entertainment and culture Shirai et al. (2007); Rofougaran et al. (2018); Lau & Chan (2008); Bourached & Cann (2019); Bourached et al. (2021); Cann et al. (2021); Stork et al. (2021); Kell et al. (2022), and security Kim & Paik (2010); Ma et al. (2018); Grant et al. (2019).

The complex, compositional character of human motion amplifies the kinematic differences between teleologically identical actions while attenuating those between actions differing in their goals. Moreover, few *real-world* tasks restrict the plausible repertoire to a small number of classes—distinct or otherwise—that could be explicitly learned. Rather, any action may be drawn from a great diversity of possibilities—both kinematic and teleological—that shape the characteristics of the underlying movements. This has two crucial implications. First, any modelling approach that lacks awareness of the full space of motion possibilities will be vulnerable to poor generalization and brittle performance in the face of kinematic anomalies. Second, the relations between different actions and their kinematic signatures are plausibly determinable only across the entire domain of action.

These considerations identify the modelling of human motion as yet another domain of machine learning where generative modelling is necessary for increased data-efficiency, generalization, and robustness. Yet, limited general purpose unsupervised methods have been investigated.

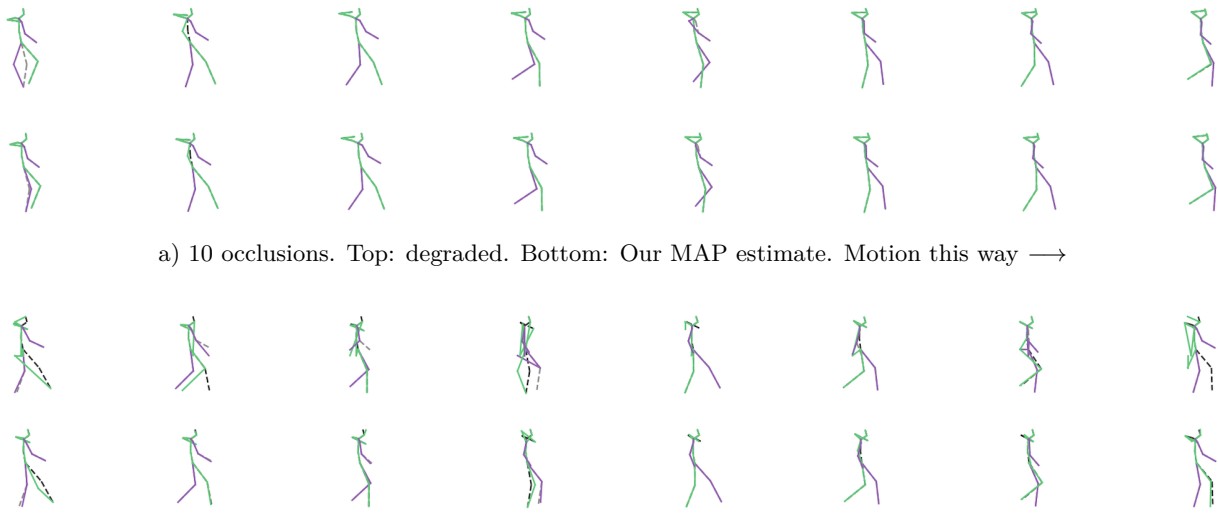

a) 10 occlusions. Top: degraded. Bottom: Our MAP estimate. Motion this way $\longrightarrow$

b) 100 occlusions. Top: degraded. Bottom: Our MAP estimate. Motion this way $\longrightarrow$

Figure 1: Motion sequence across 64 timepoints on H3.6M, sampled every 8th timepoint. Ground truth pose represented by a dotted line. The left side of the body is green, and the right is purple, while the ground truth is a dotted line where it differs.

We propose a novel architecture based on hierarchical variational autoencoders and deep graph convolutional neural networks for the application of holistic modelling of action. Our model has 4 stochastic layers $(z_0, z_1, z_2, z_3)$ which model local activity at the bottom $z_3$, which is dependent on successively more global patterns for higher latent variables, $z_{i<3}$. We create this hierarchy of abstraction by reducing graph size, via graph convolutions, for the higher latent variables. $z_0$ represents a single node – a completely global feature space that we show can effectively encode action category for conditional generation and interpretation. Our framework allows easy manipulation of stochastic depth, enabling us to investigate the effect of stochastic depth on performance. Our contributions may be summarized as follows:

1. We propose a hierarchical graph-convolutional variational autoencoder (HG-VAE) for deep generative modelling of graph structured frequencies for general purpose application in human motion modelling.

2. We demonstrate HG-VAE's ability to detect anomalies, and impute missing human motion with maximum a posteriori (MAP) estimates by gradient ascent of the model's posterior, as well as the effect this may have on downstream prediction.

3. We show that scaling to greater stochastic depth yields better likelihoods independent of model capacity.

4. We demonstrate that the HG-VAE learns an efficient hierarchical ordering, including a completely global, or abstract, representation of motion that can facilitate action-wise conditional generation and classification.

5. We provide an open-source implementation of the model we describe, available at https://anonymous.4open.science/r/generative_imputation-31EC/README.md.

## 2 Related Work

**Discriminative tasks in of human motion prediction:** Prediction and action classification are two of the main discriminative tasks in human motion modelling. Both literatures adopt similar architectures.

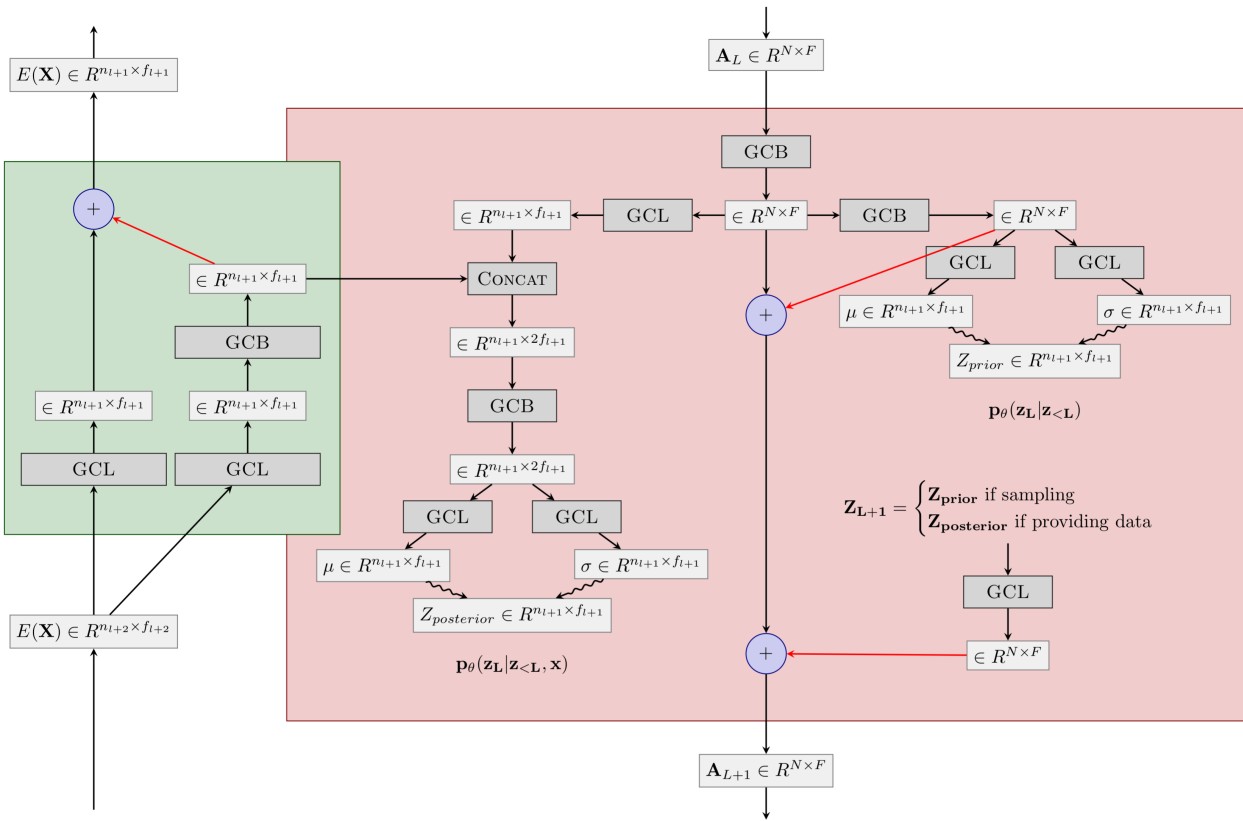

Figure 2: **A diagram of one stochastic layer, L, of our HG-VAE.** The encoder is in green, and the decoder is in red. The computational blocks GCL, and GCB correspond to equations 4, and 5 respectively. The residual connections use a learnable residual weighting Bachlechner et al. (2020) initialised to 0, where the red arrow indicates the connection weighted by the learned parameter. $N$ is the number of graph nodes in the observable, $\mathbf{x}$, and $F$ is the number of features maintained in the deterministic part of the decoder, which was 256 for all experiments. $n_l$, and $f_l$ are the number of nodes and features for the latent variable $z_l$, at stochastic layer, $L = l$.

Sequence-to-sequence prediction using Recurrent Neural Networks (RNNs) were the de facto standard for human motion prediction Fragkiadaki et al. (2015); Jain et al. (2016); Martinez et al. (2017); Pavllo et al. (2018); Gui et al. (2018a); Guo & Choi (2019); Gopalakrishnan et al. (2019); Li et al. (2020a). However, the current state-of-the-art is dominated by feed forward models Butepage et al. (2017); Li et al. (2018); Mao et al. (2019); Wei et al. (2020). Though inherently faster and easier to train than RNNs, there is little exploration of the behaviour of such models in the context of occluded or otherwise corrupted data.

**Generative modelling of human motion.** Recently there has been a trend recognising that the diversity of plausible motion necessitates the use of latent variables and thereby a distribution over synthesised or predicted motion. VAEs have been a common choice for this task: Rempe et al. (2021) use a dynamic VAE to predict a distribution over joint positions for the next pose. This approach alleviates the issue of the diversity in prediction but may still create semantically implausible continuations of motion. Or, indeed, physically impossible poses. Zhang et al. (2021) mitigate this concern by applying the SMPL-X body model at each step to project the predictions back onto the space of valid positions. While Aliakbarian et al. (2021) condition on the posterior of past observation which acts as a prior on the data posterior, thereby encouraging the latent space to carry relevant information. Mao et al. (2022), use a VAE conditioned on an action class label and show that this approach facilitates smoother transitions between actions since training data can not realistically have sufficiently diverse action transitions and lengths. Bourached et al. (2022) use a VAE to regularise discriminative models and thereby increase their robustness to out-of-distribution, yet feasible, actions.

Although these approaches address the important issue of diversity, robustness, and plausibility in the prediction of motion sequences, there are a myriad of other significant tasks that should be addressed by generative modelling: GANs are used in Wang et al. (2021) to create a plausible and diverse motion set conditioned on initial trajectory and the physical environment. Wang et al. (2021), building upon convolutional sequence generation networks proposed in Yan et al. (2019). These generative approaches to human motion have target applications in augmented reality and 3D character animations. Other GAN-based approaches such as Barsoum et al. (2018); Cai et al. (2018), are also synthetic and not designed to handle missing, or degraded data. Tevet et al. (2022), creates a diffusion based model with transformers for task-to-motion and action-to-motion synthesis. Kolotouros et al. (2021) develops a normalizing flow model to create a plausible set of predictions of 3D motion, from 2D poses. While Aliakbarian et al. (2022) also uses a flow based model to impute missing data and create realistic avatar poses while observing that real world observations have sparsity due to occlusions or other types of degradation.

The diversity of applications of latent space models for human motion highlighted here motivates the investigation of the optimal architecture for its holistic modelling. Motion-VAE Ling et al. (2020), and Hierarchical Motion VAE (HM-VAE) Li et al. (2021) aim to holistically model motion using VAEs. Specifically, HM-VAE uses a 2-layered hierarchical VAE to learn complex human motions independent of task. The bottom latent variable is a graphical representation of pose, where each feature (each joint, or their coordinates) is a node. While the top latent variable is a reduced node representation obtained by pooling adjacent joints in the encoder and unpooling in the decoder.

In this study, we define a generative mechanism that models the joint distribution over latent and observed variables and hence provides a mathematically principled machine for anomaly detection, and imputation, as well as action generation. In contrast to HM-VAE we propose an architecture that generalises to $N$ stochastic layers. We connect all latent variables by a deterministic pathway and efficiently scale depth by using rezero residual connections Bachlechner et al. (2020). Further, our model explicitly learns the contraction of joints for the higher latent variables—rather than pooling, and the top latent variable is represented by a single node providing complete abstraction in which we show action types may form well defined domains. We show that this HG-VAE yields a posterior that outperforms baselines, and provides a powerful model for imputation—therein better downstream task performance—, generation and anomaly detection.

**Graph convolutions:** Graph neural networks Kipf & Welling (2016) have received increasing attention in recent years. *Spatial graph convolutions* directly operate on vertices and their neighbors Niepert et al. (2016), and are rapidly becoming a popular tool for capturing the spatial structure of skeletons in human motion

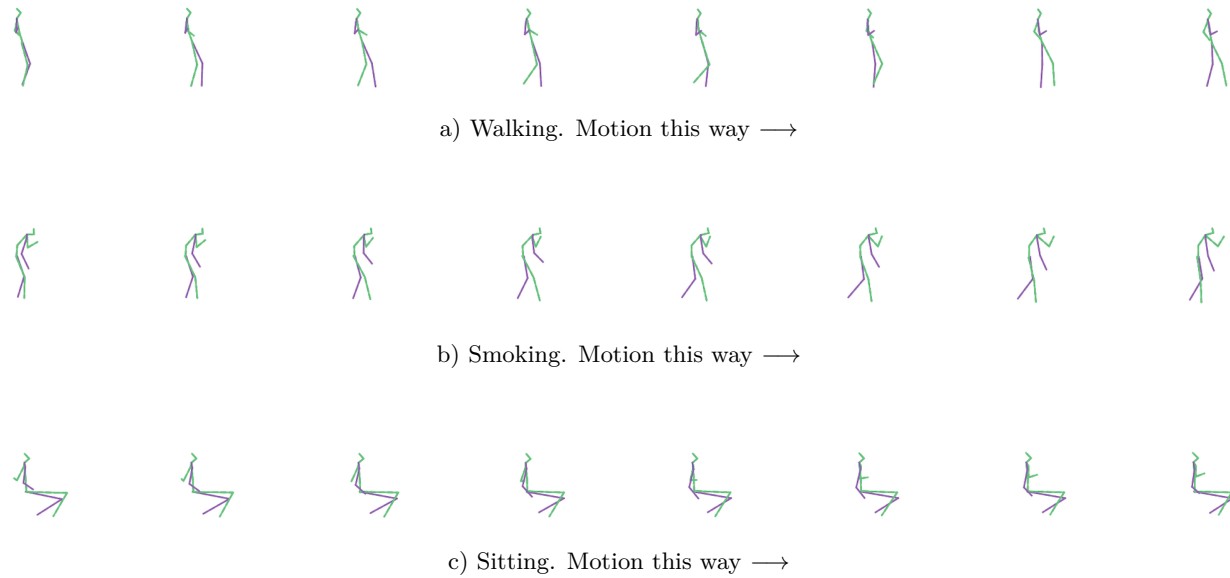

a) Walking. Motion this way $\longrightarrow$

b) Smoking. Motion this way $\longrightarrow$

c) Sitting. Motion this way $\longrightarrow$

Figure 3: Conditional samples from HG-VAE when trained conditioned on actions from H3.6M. The generated action is controlled by a one-hot vector appended to the top latent variable, $z_0 \in \mathbb{R}^{256}$. We trained on 13 actions for H3.6M, making $z_0 \in \mathbb{R}^{269}$. The left side of the body is green, and the right is purple.

Yan et al. (2018); Wang et al. (2021); Yan et al. (2019); Mao et al. (2019); Wei et al. (2020); Bourached et al. (2022); Li et al. (2020a). In particular, *spatial graph convolutions* provide a natural means of learning contractions and expansions of the number of nodes in the graph. In this work, we use this property of graph convolutions to capture, in a generative fashion, the spatial structure of skeletons at a hierarchy of graphical resolutions; from a node for each Cartesian dimension of each joint (at $z_{N-1}$), to a single node representing the global properties of the motion sequence (at $z_0$). Further, graph convolutions enable us to scale to much greater stochastic and deterministic depth while keeping the increase in the number of parameters minimal.

## 3 Preliminaries

We review prior work and introduce some of the basic terminology used in the field.

### 3.1 Variational Autoencoders

Variational AutoEncoders (VAEs) Kingma & Welling (2013) consist of a generator $p_\theta(x|z)$, a prior $p(z)$, and an approximate posterior $q_\phi(z|x)$. Neural networks parametrized by $\phi$ and $\theta$ are trained end-to-end with backpropagation and the reparameterization trick in order to maximize the evidence lower bound (ELBO)

$$\log p_\theta(\mathbf{x}) \geq \ \mathbb{E}_{\mathbf{z} \sim q_\phi(\mathbf{z}|\mathbf{x})} \log p_\theta(\mathbf{x} \mid \mathbf{z}) \ - D_{\mathrm{KL}} \left[ q_\phi(\mathbf{z} \mid \mathbf{x}) \| p_\theta(\mathbf{z}) \right]. \tag{1}$$

### 3.2 Hierarchical Variational Autoencoders

Originally, VAEs used fully factorised Gaussians for the prior, $p(z)$, and the approximate posterior, $p_\phi(z|x)$. This can induce the generated, $p_\theta(x|z)$, to have the property that correlated features blur together, this is because the interpolation of any two instances in the latent variable, $z$, is continuous. However, most complex distributions contain variables that have dependency on other variables.

In Sønderby et al. (2016), a Hierarchical VAE (HVAE) structure, also called a *ladder*, or *top-down* VAE is proposed, where the prior and posterior generate latent variables as

$$p_\theta(\boldsymbol{z}) = p_\theta(\boldsymbol{z}_0) \, p_\theta(\boldsymbol{z}_1 \mid \boldsymbol{z}_0) \ldots p_\theta(\boldsymbol{z}_N \mid \boldsymbol{z}_{<N}), \tag{2}$$

$$q_\phi(\boldsymbol{z} \mid \boldsymbol{x}) = q_\phi(\boldsymbol{z}_0 \mid \boldsymbol{x}) \, q_\phi(\boldsymbol{z}_1 \mid \boldsymbol{z}_0, \boldsymbol{x}) \ldots q_\phi(\boldsymbol{z}_N \mid \boldsymbol{z}_{<N}, \boldsymbol{x}). \tag{3}$$

In this structure, $\phi$ first performs a deterministic *bottom-up* pass, producing multiple sets of features of $x$, $\{f_N(x), f_{N-1}(x), \cdots, f_1(x)\}$. Then $\theta$ performs a *bottom-down* pass producing latent variables from $i = 0$, to $i = N - 1$ as: $p_\theta(z_i)$, and $q_\phi(z_i | f_i(x))$. Child (2021) shows that this structure actually generalises autoregressive models—models that probabilistically select states or variables based on previous states or variables—, and with the right training conditions and architectural choices, will outperform them across a range of major vision datasets.

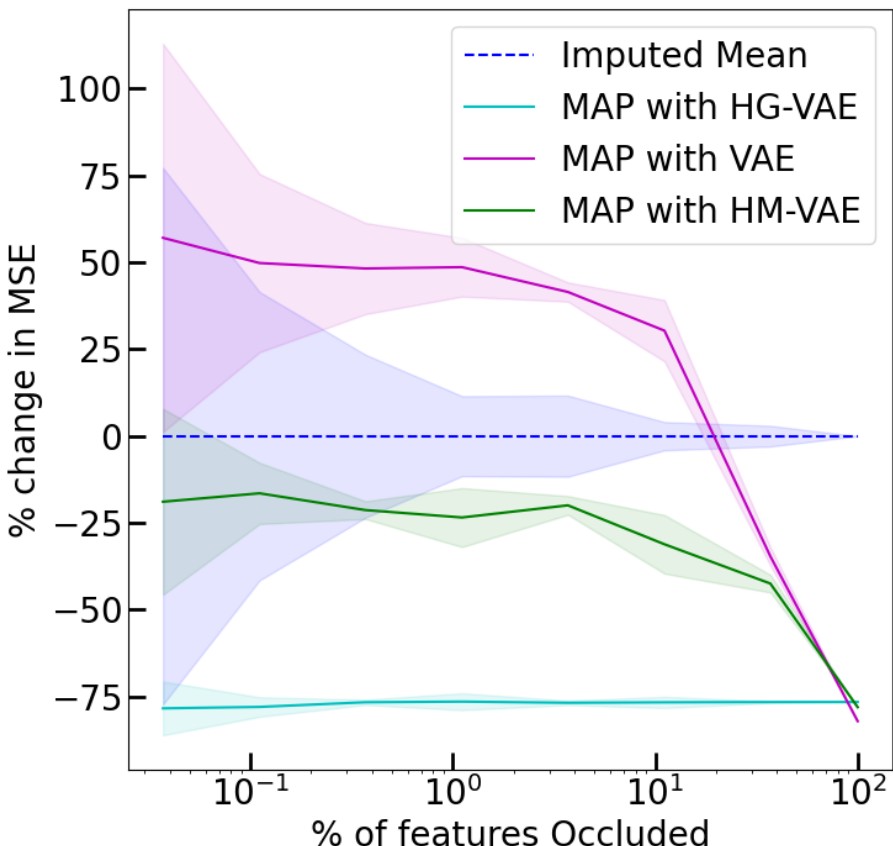

Figure 4: Percentage change in MSE from the ground truth of the MAP estimates with respect to mean imputation as a function of percentage of features occluded. HG-VAE gives a $77\% \pm 1\%$ decrease from mean imputation consistently across all degrees of degradation.

## 4 Our Approach

We design a hierachical graph-convolutional variational autoencoder (HG-VAE), that avails of hierarchical latent variables with a graph-convolutional structure. Mao et al. (2019); Wei et al. (2020); Bourached et al. (2022); Zhang et al. (2021) use temporal frequencies encoded by DCT as features of nodes in a graph on which graph-convolutional operations are performed. This is similar to how Convolutional Neural Networks, for vision tasks Lawrence et al. (1997); Child (2021), expand RGB channels into a larger set of feature channels

while preserving a 2-dimensional spatial structure through a spatial-convolutional operation. Early work with Graph Convolutional Networks (GCNs) used a symmetric (binary, or weighted) adjacency matrix such as in Kipf and Welling Kipf & Welling (2016), where the applications were citation networks, or knowledge graphs. Such methods are referred to as *spectral graph convolutions*, which operate on the spectral domain via graph Laplacians. However, usage in human motion prediction tasks necessitates a learnable weighted adjacency matrix, known as *spatial graph convolutions* which, for convenience, we'll refer to in short as a graph convolution. Li et al. (2020b) show that representing the skeletal graph at multiple scales is a sensible representation for discriminative purposes. Further, Yan et al. (2019); Wang et al. (2021) synthesise motion by sampling latent variables of successively greater graph resolution. In juxtaposition to Mao et al. (2019); Wei et al. (2020); Bourached et al. (2022); Li et al. (2020b), which have a discriminative focus, our objective is to obtain latent variables of a hierarchical structure. We use different hierarchical levels to represent variables at different graphical scales, and levels of abstraction.

We refer to two main neural network computational blocks as a Graph Convolutional Layer (GCL), which takes as input a graph $A \in \mathbb{R}^{N_{in} \times F_{in}}$ with number of nodes, $N_{in}$, and number of features, $F_{in}$, and outputs a new graph with number of nodes, $N_{out}$, and number of features, $F_{out}$. $F$ may be considered much like feature channels in a Convolutional Neural Network. For this problem they represent features derived from the frequencies of motion. The second main neural network computational block we refer to as a Graph Convolutional Block (GCB) which takes as input graph $A \in \mathbb{R}^{N_{in} \times F_{in}}$, and outputs a graph of the same dimensionality. Mathematically, these are defined as

$$\text{GCL}(\mathbf{A}) = \sigma\Big(\mathbf{SAW} + \mathbf{b}\Big) \tag{4}$$

$$\text{GCB}(\mathbf{A}) = \sigma\Big(\mathbf{S_2}\sigma\Big(\mathbf{S_1AW_1} + \mathbf{b_1}\Big)\mathbf{W_2} + \mathbf{b_2}\Big) + \alpha\mathbf{A}, \tag{5}$$

where in equation 4, $\mathbf{S} \in \mathbb{R}^{N_{out} \times N_{in}}$, $\mathbf{W} \in \mathbb{R}^{F_{in} \times F_{out}}$, and $\mathbf{b} \in \mathbb{R}^{N_{out} \times F_{out}}$ are all learnable parameters. Similarly, $\mathbf{S}, \mathbf{W}, \mathbf{b} \in \mathbb{R}^{N_{in} \times F_{in}}$, in equation 5, are also learnable parameters. $\alpha$ is a learnable residual weighting that is initialised to 0 as proposed recently by Bachlechner et al. (2020) as an efficient alternative to batch normalization, Ioffe & Szegedy (2015), to mitigate the effect of covariate shift. $\sigma(\cdot)$ is a Gaussian Error Linear Unit, Hendrycks & Gimpel (2016), (GeLU).

## 5 Implementation

**Network Architecture:** In the top down decoders we maintain a principal deterministic route connected to each latent variable such that latent variables may contribute directly to the output without being required to contribute to composite features in dependent variables. This approach increases the speed and stability of training by simplifying the nature of the causal relationships between the latent variables and $\mathbf{x}$, where possible. All computational blocks are of the form of either equation 4 or equation 5. The encoder-decoder architecture for a single stochastic level is shown in Figure 2.

**Training Details:** We use the ELBO, defined by equation 1, to train the network end-to-end using the ADAM optimizer Kingma & Ba (2014), with a learning rate of 0.001 and a batch size of 800. We set $p_\theta(\mathbf{x}|\mathbf{z}) = \mathcal{N}(\mu(\mathbf{z}), \sigma(\mathbf{z}))$, where $\mu$ and $\sigma$ are outputs of the final stochastic decoder. On the H3.6M dataset, we appended a one-hot vector to $z_0$ to represent each action class.

We use two main techniques for increasing training stability; gradient clipping, enforcing a gradient norm value of 100.0, and a similar warm-up procedure for the Kullback-Leibler ($D_{KL}$) divergence to Sønderby et al. (2016), and Child (2021), increasing its weighting in the loss linearly from 0.001 to 1.0 over the first 200 epochs. We train for a total of 5000 epochs, which requires ca. one week on a NVIDIA GeForce RTX 2070 GPU.

# 6 Experiments

## 6.1 Datasets

We are given a motion sequence $\mathbf{X}_{1:N} = (\mathbf{x}_1, \mathbf{x}_2, \mathbf{x}_3, \cdots, \mathbf{x}_N)$ consisting of $N$ consecutive human poses, where $\mathbf{x}_i \in \mathbb{R}^K$, with $K$ the number of parameters describing each pose. The temporal components are converted into frequencies using the Discrete Cosine Transformation (DCT) prior to input into the network, and then back to timepoints using the Inverse Discrete Cosine Transformation (IDCT) before being applied to the loss. Further details are supplied in the appendix.

**AMASS:** The Archive of Motion Capture as Surface Shapes (AMASS) dataset Mahmood et al. (2019) is the largest open-source human motion dataset, which aggregates a number of mocap datasets, such as CMU, KIT and BMLrub, using a SMPL Loper et al. (2015); Romero et al. (2017) parametrization to obtain a human mesh. SMPL represents a human by a shape vector and joint rotation angles. The shape vector, which encompasses coefficients of different human shape bases, defines the human skeleton. We obtain human poses in 3D by applying forward kinematics to one human skeleton. In AMASS, a human pose is represented by 52 joints, including 22 body joints and 30 hand joints. Since we are interested in skeletal motion, we follow Wei et al. (2020) and discard the hand joints and the 4 static joints, leading to an 18-joint human pose. Further, we also use BMLrub[1] (522 min. video sequence), as our test set as each sequence consists of one actor performing one type of action. All quantitative results are reported from this dataset. We consider motion sequences of 50 timepoints taking only every second frame. A single datapoint here hence consists of 54 nodes, and 50 timepoints. Forming 2700 dimensional inputs.

**Human3.6M (H3.6M):** The H3.6M dataset Ionescu et al. (2011; 2013), so called as it contains a selection of 3.6 million 3D human poses and corresponding images, consists of seven actors each performing 15 actions, such as walking, eating, discussion, sitting, and talking on the phone. Martinez et al. (2017); Mao et al. (2019); Li et al. (2020a) all follow the same training and evaluation procedure: training their motion prediction model on 6 (5 for train and 1 for cross-validation) of the actors, and use subject 5 as a heldout test set. In this work, we use our model trained on H3.6M to demonstrate most of our qualitative results.

## 6.2 Baselines

**VAE:** A conventional, fully-connected, VAE is trained with $n_z = 50$ dimensional latent variable and a symmetric encoder-decoder of architecture $2000, 1000, 500, 100, 50$, with batch normalization applied to each layer and trained for 200 epochs with a learning rate of 0.001 using the ADAM optimizer. The model has 20.81M parameters.

**HM-VAE:** We implement and train the motion prior model proposed by Li et al. (2021). We pool the joints in the upper latent variable to 6 joints, which results in a graph size of 18 at the top latent variable, and 54 on the bottom. In our model, we directly model the scale of the uncertainty in the reconstruction, $\sigma(z)$, which implicitly learns an appropriate scaling between the two terms in the ELBO (equation 1). However, for HM-VAE we downweight the $D_{\mathrm{KL}}$ divergence by a factor of 0.003, as in Li et al. (2018). We otherwise use the same hyperparameters as for our model.

## 6.3 Experimental setup

We present experiments in support of 3 main arguments for the proficiency of our generative model. First, we degrade the test set data by simulating occlusions wherein, a naïve imputation, the mean value of the missing feature over the train set, is substituted for the ground truth. The degree of degradation is controlled by the number of input features occluded. A sufficient level of degradation may be considered as shifting the data out-of-distribution, which we quantify using the model's posterior. We show that this well-informed imputation facilitates better downstream discriminative tasks by considering the degradation in performance of two deterministic deep network motion prediction models, convSeq2Seq Li et al. (2018), and HisRepItself

---

[1] Available at `https://amass.is.tue.mpg.de/dataset`.

Wei et al. (2020) under feature occlusion. We train each model on 50 timepoints, to predict the next 25. At test time we randomly occlude the input features and report the error on the predicted future trajectory. At test time we randomly occlude the input features and report the error on the predicted future trajectory. We report the Mean Per Joint Position Error (MPJPE) Ionescu et al. (2013) given by

$$\ell_m = \frac{1}{J(N+T)} \sum_{n=1}^{N+T} \sum_{j=1}^{J} \|\hat{\mathbf{p}}_{j,n} - \mathbf{p}_{j,n}\|^2 \tag{6}$$

where $\hat{\mathbf{p}}_{j,n} \in \mathbb{R}^3$ denotes the predicted jth joint position in frame $n$. And $\mathbf{p}_{j,n}$ is the corresponding ground truth, while J is the number of joints in the skeleton.

Second, we examine the effect of stochastic depth, independent of capacity, on performance by training a 16 stochastic-layered model several times with a varying number of its top latent variables *turned off* ($z_L = \mu_L$, for each level, $L$, that is *turned off*, and $KL_L$ does not contribute to the loss).

Third, we demonstrate that HG-VAE learns an efficient hierarchical ordering that facilitates levels of abstraction, the top of which may represent action classification as well as conditional generation.

## 7 Results

Computing the average posterior on the H3.6M test set yielded the value $p(z|x) = -9312$ with a standard deviation of 500. This is the average negative log probability predicted by our trained model. We use this as a benchmark to compare anomalies, below. Figure 1, a and b, show two motion sequences (top of each) with the $p(z|x) = -10039$, and $p(z|x) = -10364$ respectively, meaning that both are well outside a standard deviation of the average. The bottom row of a and b show the MAP-imputed values using HG-VAE, with $p(z|x) = -9643$, and $p(z|x) = -9717$ respectively. A reasonable out-of-distribution (OoD) threshold of the standard deviation of the posterior would both enable the model to detect these OoD examples, and, if the samples are OoD due to occlusion, find the most probable in-distribution representation.

In figure 5 we compare the raw performance of convSeq2Seq, and HisRepItself, given mean imputation for occlusions, against the MAP estimates obtained by performing 100 steps of gradient ascent on HG-VAE's posterior, as well as on HM-VAE's posterior. The use of MAP estimates from our model give significant improvement in downstream prediction task compared to HM-VAE, as measured by MPJPE indicating the greater quality of $p(z|x)$, and capacity of our model for handling complex non-linear relationships present in human motion.

In Figure 4 we consider the MSE between the ground truth and the naïve mean imputation; as well as each models' MAP estimates. We show this as a percentage difference in MSE compared to the MSE of mean imputation. HG-VAE gives a 77%±1% decrease consistently across all degrees of degradation, which greatly beats baseline methods.

We investigated how statistical depth, independent of capacity, improved performance. Here a stochastic depth of 1 is a classical VAE with a very deep encoder, and decoder. Simply, $z_L = \mu_L$, for each level $L$ that was *turned off*, and $KL_L$ didn't contribute to the loss. Table 1 shows that greater stochastic depth achieves higher likelihoods, reconstructions, and KL. Implying that the model benefits greatly from the latent variable dependency structure.

Figure 3 show samples from the model trained on H3.6M with one-hot labels for walking, smoking, and sitting. Each action clearly emulates its purported action class indicating the spatial abstraction of the top latent variable achieved by the learned contractions.

To further explain this, Figure 6 shows a conditional sample drawn with the label *walking* from a 4-layer HG-VAE at each level of resolution. 6a is the mean sample drawn from the walking cluster (see appendix), $Z_i = \mu_i$, for $i > 0$. While 6b draws genuine sample at the top level, $Z_0 = z_0$, and the mean for lower levels, $Z_i = \mu_i$, for $i > 1$, etc. We can see that each latent level adds further expression to the motion sequence,

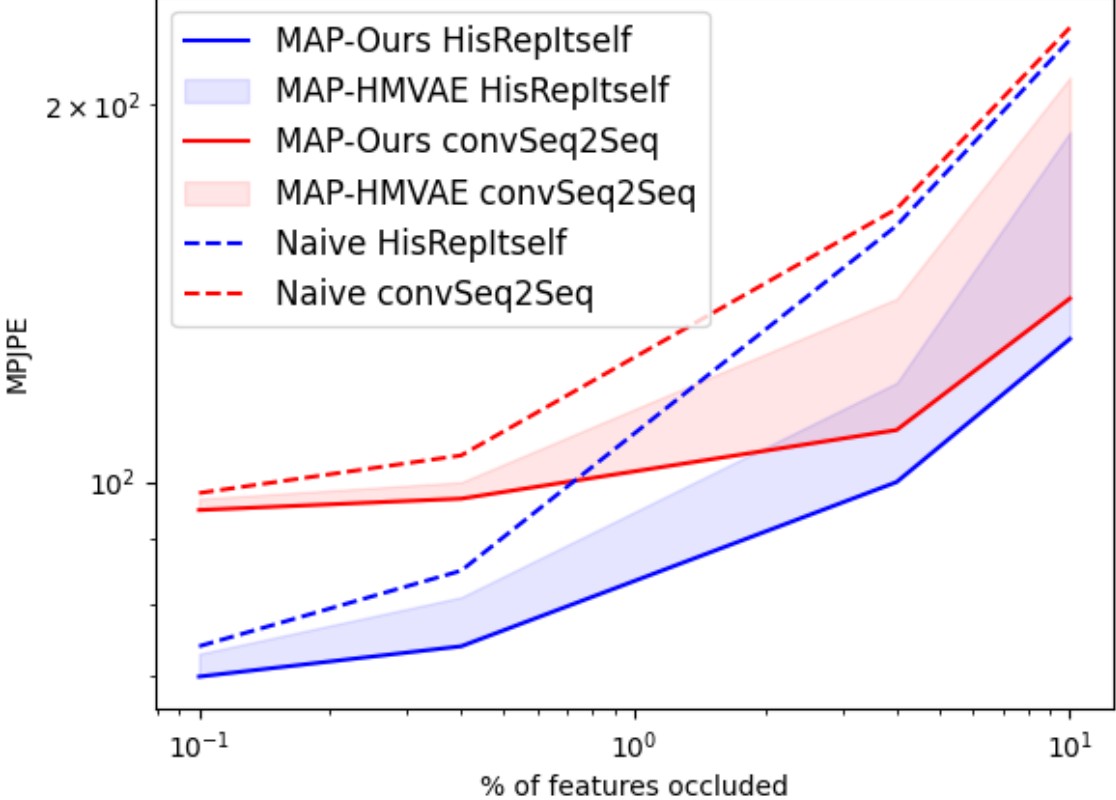

Figure 5: MPJPE for discriminative prediction models with varying degree of degradation with and without including the MAP estimate of HG-VAE and HM-VAE on the occluded features.

Table 1: HG-VAE Performance as a Function of Stochastic Depth.

| Parameters | Stochastic Depth | $\log(\mathbf{X})$ | MSE | KL-Divergence |
|---|---|---|---|---|
| 15M | 1 | 1986 | 2.30 | 274 |
| 15M | 2 | 3912 | 1.31 | 79 |
| 15M | 4 | 6511 | 0.59 | 30 |
| 15M | 8 | 7112 | 0.55 | 32 |
| 15M | 16 | 7238 | 0.53 | 29 |

illustrating that low levels of the hierarchy (6d) model fine movements, and local relationships, while higher levels model a much more course 'snapshot' of the motion sequence.

## 8 Conclusions

We propose a novel hierarchical graph-convolutional variational autoencoder suited to the diversity and complexity of human motion modelling and demonstrate its ability to learn complex and action-generic latent distributions for human motion that may be used for highly informed imputation and anomaly detection, as well as downstream discriminative tasks. We show that with just 10 gradient ascent steps we obtain a 77% decrease in MSE compared to a mean imputation policy consistently across all degrees of degradation which is greater and more consistent than all baseline models. We demonstrate that stochastic depth matters,

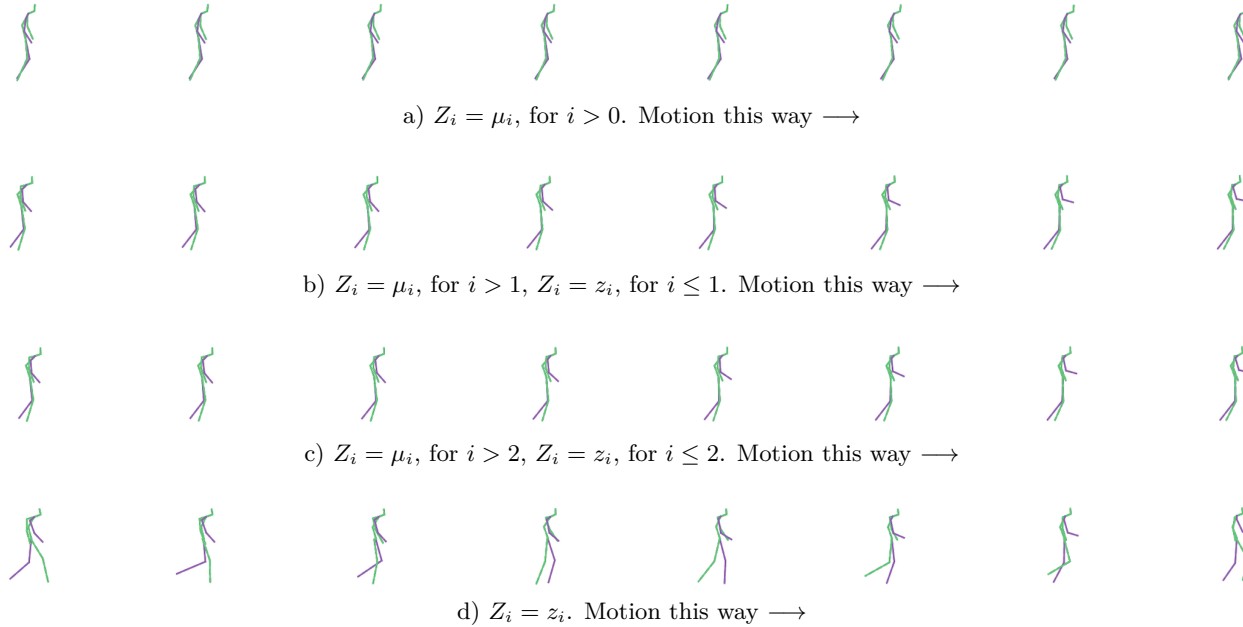

a) $Z_i = \mu_i$, for $i > 0$. Motion this way $\longrightarrow$

b) $Z_i = \mu_i$, for $i > 1$, $Z_i = z_i$, for $i \leq 1$. Motion this way $\longrightarrow$

c) $Z_i = \mu_i$, for $i > 2$, $Z_i = z_i$, for $i \leq 2$. Motion this way $\longrightarrow$

d) $Z_i = z_i$. Motion this way $\longrightarrow$

Figure 6: Motion from left to right. A walking sample from a 4-layered HG-VAE. Given $Z_0 = z_0$, $p(z)$ was generated as in equation 2. We extract only the mean for each $Z_{>i}$.

independent of model capacity. We further show that the hierarchical latent variables have both enough expression in the lower latent variables ($z_{i>0}$) to model local observables while benefiting from complete abstraction in the top latent variable ($z_0$) such that it may be class-conditioned by appending a one-hot vector and produce qualitatively distinct actions. Due to the generality of the architecture, HG-VAE may be a suitable, deeply expressive, generative model for many applications in the modelling of human motion.

**Broader Impact Statement**

The method of missing data imputation via gradient ascent on the posterior, though powerful given a good generative model, is expensive. Despite only a small number of parameters—the number of missing features—each step needs almost a complete forward and backward pass through the model. So this may not be an effective method of imputation for online models. However, using a small number of posterior ascent steps is shown to already yield significant improvement, and may be applicable for many online applications. Furthermore, such a model trained on a motion capture dataset such as H3.6M, or AMASS—as is the case here—may be used to very effectively improve motion capture data containing occlusions that is captured in the wild by pose estimation methods. Further work might also include using this model trained on AMASS to synthesise larger datasets. We acknowledge that models capable of learning highly specific motion patterns could be used for nefarious, Orwellian, applications. However, we believe the potential benefit in an ever more automated world, especially in safety in healthcare, outweighs the potential cost.

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

## Appendix

The appendix consists of 2 parts. We provide a brief summary of each section below.

Appendix A: we provide an elaboration of the formulation of the data and the equations used to transform the temporal component of the data before input to the model, as well as after output.

Appendix B: we provide further results and explanation of the imputation experiments available in the main text.

## A  Problem formulation

### A.1  DCT-based Temporal Encoding

Mao et al. (2019) proposed transforming the temporal component of human motion to frequencies using Discrete Cosine Transformations (DCT). In this way each resulting coefficient encodes information of the entire sequence at a particular temporal frequency. Furthermore, the option to remove high or low frequencies is provided. Given a joint, $k$, the position of $k$ over $N$ time steps is given by the trajectory vector: $\mathbf{x}_k = [x_{k,1}, \ldots, x_{k,N}]$ where we convert to a DCT vector of the form: $\mathbf{C}_k = [C_{k,1}, \ldots, C_{k,N}]$ where $C_{k,l}$ represents the lth DCT coefficient. For $\delta_{l1} \in \mathbb{R}^N = [1, 0, \cdots, 0]$, these coefficients may be computed as

$$C_{k,l} = \sqrt{\frac{2}{N}} \sum_{n=1}^{N} x_{k,n} \frac{1}{\sqrt{1 + \delta_{l1}}} \cos\left(\frac{\pi}{2N}(2n-1)(l-1)\right). \tag{7}$$

If no frequencies are cropped, the DCT is invertible via the Inverse Discrete Cosine Transform (IDCT):

$$x_{k,l} = \sqrt{\frac{2}{N}} \sum_{l=1}^{N} C_{k,l} \frac{1}{\sqrt{1 + \delta_{l1}}} \cos\left(\frac{\pi}{2N}(2n-1)(l-1)\right). \tag{8}$$

We find this lossless transformation effectively creates a feature space that is much more conducive for optimisation. The objective is still to learn $p(x)$.

## B  Imputation

Figure 7 shows the initial posterior of the degraded input as well as the final after 10 steps, a $p(z|x)_{\text{MAP}}$ estimate, of gradient ascent on the model's posterior. For the HG-VAE (7b) the average posterior already lies outside the standard deviation of the ground truth for just 0.5% of features occluded, demonstrating acute out-of-distribution detection. In contrast, the VAE's average posterior only falls outside a standard deviation of the ground truth for approximately 50% occlusion. A fully expressive model should not, on average, give a higher MAP estimate than the posterior on the ground truth. Note, we show comparison to the VAE to illustrate this point, not to prove the efficacy of our model—which is evidenced by the experiments in the main text. Figure 7a indicates a lack of capacity to delineate latent features representative of the full expressivity of $x$, and hence gives a higher probability to a closer to average expression than the ground truth.

We see no evidence of this lack of expressivity in Figure 7b. In fact, the MAP estimate strictly upper-bounds the mean imputation, $p(z|x)_{\text{init}}$, and is strictly upper-bounded by the ground truth $(p(z|x)_{\text{GT}})$.

**Imputation of missing data:**  Given occluded inputs, and a sufficiently expressive $p(z|x)$, it is possible to gradient ascent the missing inputs on the model's posterior. If the model's maximum a posteriori (MAP), given the partially observed datapoint, is close to the ground truth, $x$, this will result in a statistically

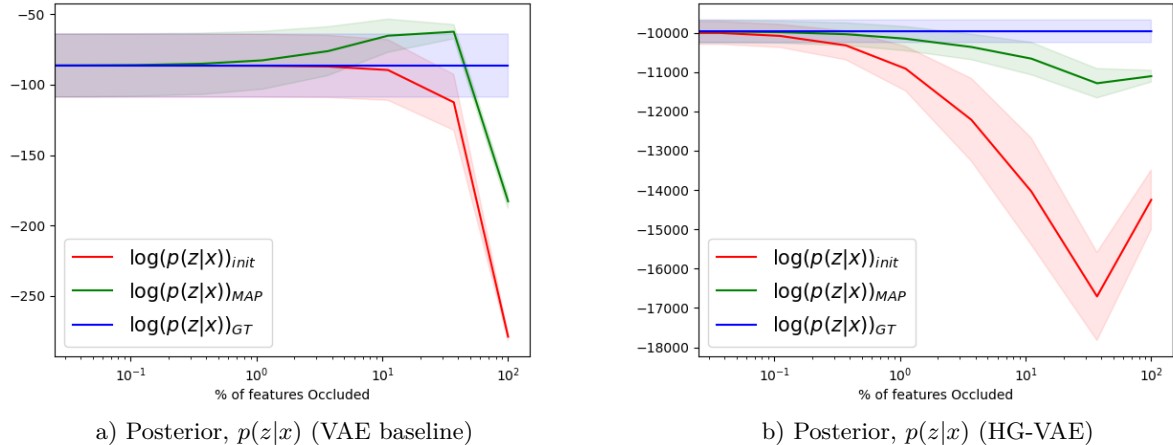

a) Posterior, $p(z|x)$ (VAE baseline)  b) Posterior, $p(z|x)$ (HG-VAE)

Figure 7: Posterior, $p(z|x)$, for the motion data of 50 timepoints on BMLrub with increasing level of degradation. Note $p(z|x)_{MAP}$ is a MAP estimate obtained via gradient ascent. Number of features occluded is out of a maximum of 2700.

well-informed imputation. For these experiments we use the Adam optimiser on the negative log posterior $\Big($ $-\sum_i \log\big(p(z_i|x)\big)\Big)$ and select a learning rate that is sufficiently low for stable descent. This was 100.0 for the VAE and 1.0 for HG-VAE, and HM-VAE (this difference is proportional to the difference in magnitude of the average log posterior for the respective models). We gradient ascended on 800 datapoints at once, for a maximum of 10 steps selecting the step of highest model posterior for each datapoint.

Figure 8 shows the results from Figure 7b as a percentage change in the negative log posterior. We can see that the change on a logarithmic scale is consistent over all degrees of degradation indicating the sensitivity of the model to subtle changes in the observables.

We use Uniform Manifold Approximation and Prediction (UMAP) McInnes et al. (2018), to project the top latent variables of 1000 random samples, conditioned on one-hot encodings, onto a 2-dimensional plane. We show two contrasting actions in figure 9, *walking*, and *sitting*. We can see that a very clear separation between the two different actions showing that there is a strong level of abstraction in the top latent variable.

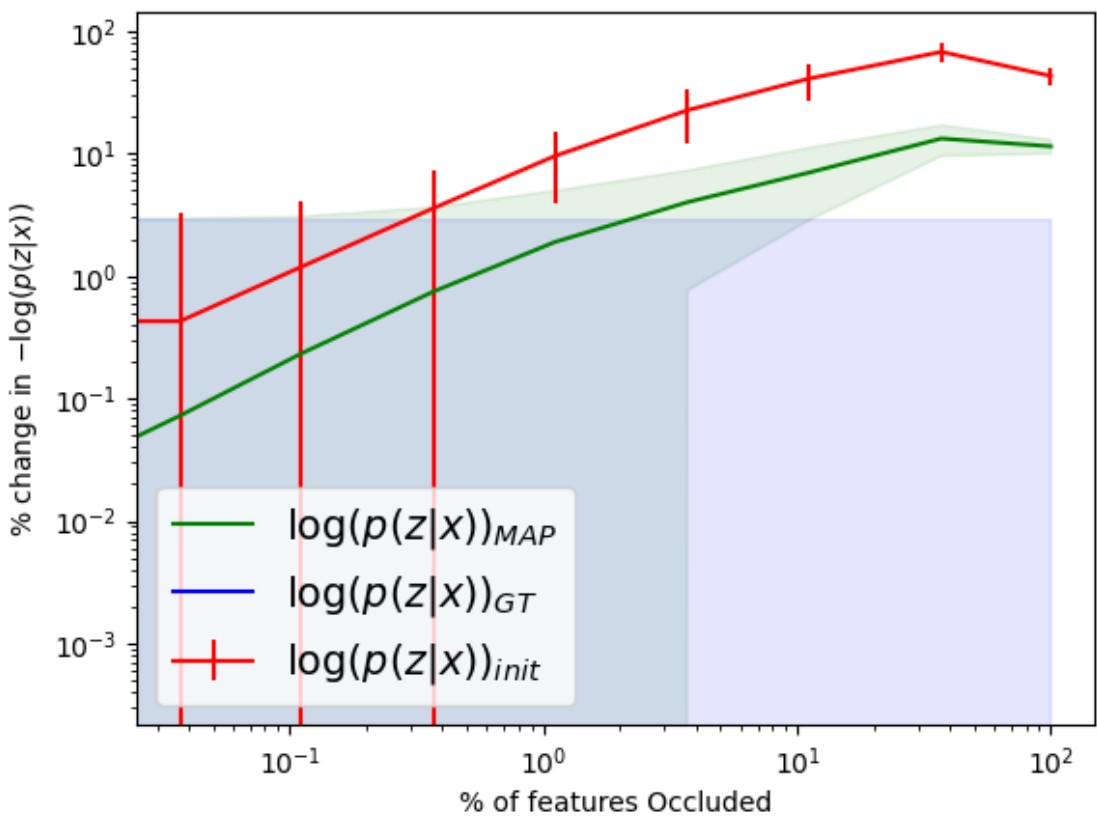

Figure 8: % change in negative log posterior in the HG-VAE for the motion data of 50 timepoints on BMLrub with increasing levels of degradation. Degree of degradation displayed on a log-log scale to illustrate the consistency of the model for small degrees of degradation.

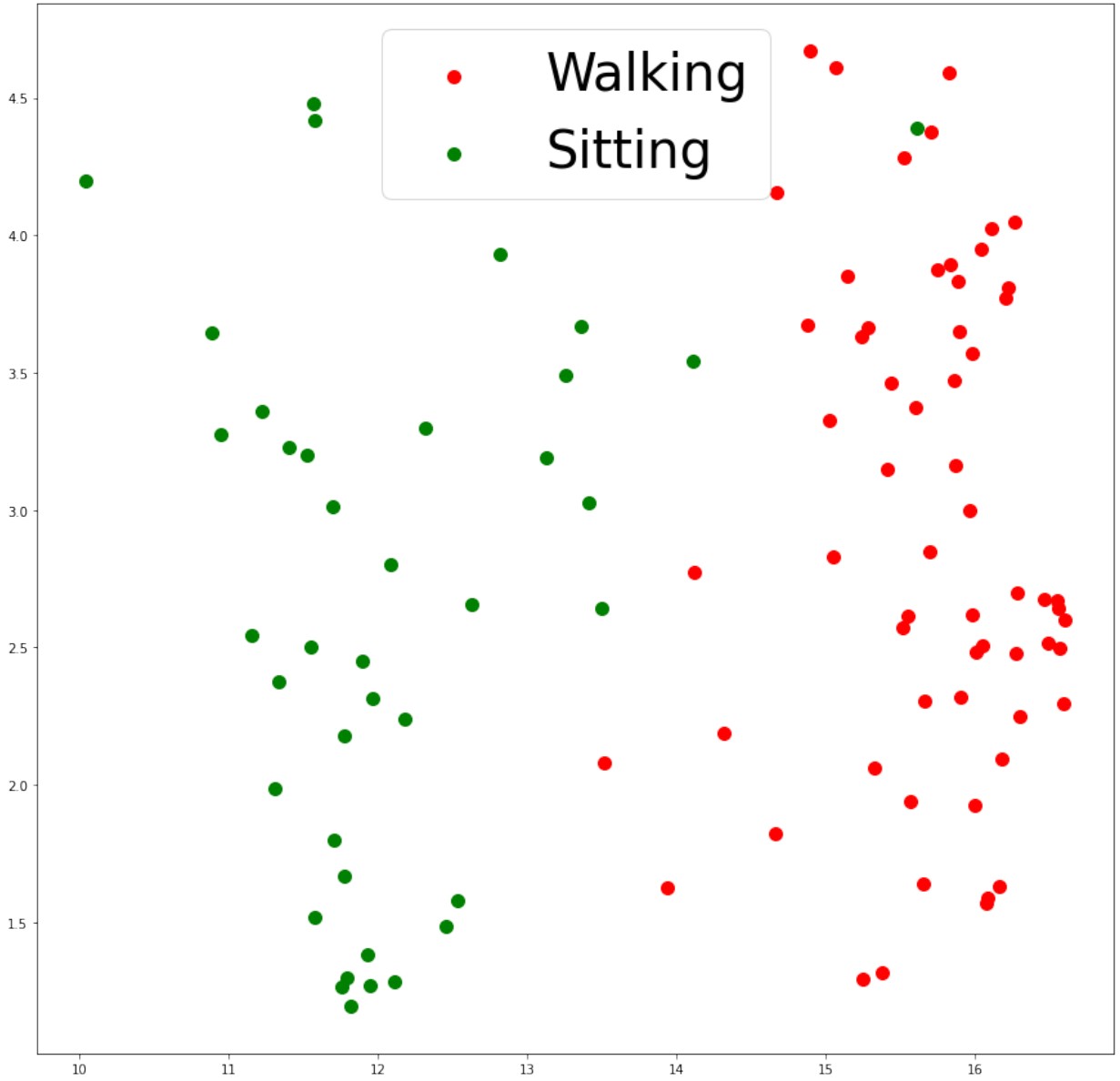

Figure 9: UMAP projection of $z_0$ for the conditional generation of two contrasting classes; *walking*, and *sitting*.

