# OpenReview forum: "Hierarchical Graph-Convolutional Variational Autoencoding for Generative Modelling of Human Motion"
_TMLR — Rejected by TMLR_

### Review · Reviewer_rL2b · 2024-03-03

**Summary Of Contributions:**

This paper present a hierarchical graph convolutional VAE to generate model of action. The proposed architecture relies on hierarchical latent representation, with 4 stochastic layers, where earlier stochastic layers global representation of action and the bottom ones represent more local activities. Authors demonstrate that more stochastic layers contributes more strongly to better likelihood estimates of the motions.

**Audience:**

No

**Broader Impact Concerns:**

No major concern

**Claims And Evidence:**

No

**Requested Changes:**

Weaknesses in the above section should be address through a major and thorough revision.

**Strengths And Weaknesses:**

**Strength**:

The paper focuses on learning the distribution of complex human motion, which indeed is a challenging and interesting problem.

**Weaknesses**:

- Paper lacks technical novelty and lacks discussion of very similar efforts in this area, making it hard to judge which component of the proposed method is novel.
- Paper lacks discussion of recent efforts in this area. The field of human motion analysis, specially with probabilistic generative models, has been growing very fast, and authors missed many important works after 2021-2022 (see references below)
- The paper lacks clarity in writing the proposed method and highlighting its differences to existing techniques. Particularly, the main idea of the paper is presented in the preliminary section! The preliminary section seems over explained, e.g., section 3.2 and 3.1 can be summarized through citations. Also, long equation in section 3.2 is not labeled correctly.
- The baselines used for comparison are not complete (see references below). Also, authors only reported results on Human3.6M, missing more challenging datasets to report on (e.g., standard AMSS test split results are not provided comprehensively in the main paper).
- Qualitative results can be improved. Illustrating stickman make the judgement of plausibility of the generated motions very hard. One potential improvement is to use a parametric model of human body (e.g., SMPL) with neutral shape parameters, allowing better illustration of the 3D pose of human. Moreover, for such tasks, authors are highly recommended to attach a supplementary video.
- Motion reconstruction quality is not studied thoroughly in this work. While some aspects of robustness to OOD is considered, many strong metrics for motion reconstruction (MPJPE, MPJVE, ...) and metrics related to quality and diversity of generated motions are not discussed. Results in section 6 are not well discussed. Authors are encouraged to make it clearer ,e.g., what does −10364 mean in `p(z|x) = −10364`. Is it good or bad? With standard deviation of 500, it is hard to judge the comparisons made in the following sentence of section 6. Overall, I encourage authors to better explain the results.
- Paper discusses OOD motions. However, the proposed method is only compared to limited VAE-based methods, failing to compare against other generative models of human motion, e.g., Diffusion models, Flow-based models, or more recent VAE-based models (see references below).


_References_ (+ many more recent works in this area, but these are the ones in similar-ish years authors used to discuss related work and make comparisons)

- Rempe, Davis, et al. "Humor: 3d human motion model for robust pose estimation." Proceedings of the IEEE/CVF international conference on computer vision. 2021.
- Kolotouros, Nikos, et al. "Probabilistic modeling for human mesh recovery." Proceedings of the IEEE/CVF international conference on computer vision. 2021.
- Aliakbarian, Sadegh, et al. "Flag: Flow-based 3d avatar generation from sparse observations." Proceedings of the IEEE/CVF Conference on Computer Vision and Pattern Recognition. 2022.
- Tevet, Guy, et al. "Human motion diffusion model." arXiv preprint arXiv:2209.14916 (2022).
- Mao, Wei, Miaomiao Liu, and Mathieu Salzmann. "Weakly-supervised action transition learning for stochastic human motion prediction." Proceedings of the IEEE/CVF Conference on Computer Vision and Pattern Recognition. 2022.
- Aliakbarian, Sadegh, et al. "Contextually plausible and diverse 3d human motion prediction." Proceedings of the IEEE/CVF International Conference on Computer Vision. 2021.
- Zhang, Yan, Michael J. Black, and Siyu Tang. "We are more than our joints: Predicting how 3d bodies move." Proceedings of the IEEE/CVF Conference on Computer Vision and Pattern Recognition. 2021.

---

> ### Author Response · Authors · 2024-03-24
> **Response, and details of submitted revised version related to this feedback.**
>
> Thank you for your comprehensive feedback and suggestions. Please see below what has been addressed in the uploaded revised version.
>
> Thank you for highlighting the missing related work. A comprehensive rewriting of the related work section now includes these works and a more clear narrative for our objective and choice of baselines, as well as the novelty.
>
> The point on the preliminary section being too long and including stuff it shouldn’t is well received. 3.3 Has been made it’s own section called ‘Our Approach’ and the content adjusted accordingly. Further, 3.2 has been reduced to a more concise explanation.
>
> All experiments, except for the action class conditional generation, were conducted on AMASS. The reference to H3.6M in the results section was with regard to the qualitative results only. This is not clear in the text and the explanation has been restructured accordingly.
>
> Stick men were chosen for the qualitative results because they were more clear for degradation experiments, and in facilitating a clear degree of error by-eye. It is appreciated that this often does not look as good SMPL, however it seemed appropriate for the experiments in this paper.
>
> The motivation for choosing HMVAE as a baseline was that it attempts to create a general purpose, holistic model of motion as the objective of the paper is to motivate HGVAE as SOTA general purpose model of human motion without training it specifically for imputation, prediction, or anomaly detection directly. However, although this point is of central important, it has not been well explained in the text and this point has been refactored into the related work, our approach, experiments, and results sections in a comprehensive revision. In particular, the results section has been thoroughly rethought-out and extended given your feedback.

---

### Review · Reviewer_hEJH · 2024-03-05

**Summary Of Contributions:**

The submitted manuscript focuses on modeling human motion via a hierarchical variational autoencoder, implemented via graph convolutional layers. The proposed HG-VAE is supposed to be able to detect out-of-distribution data, and even to handle missing data. Ablation studies and comparisons with a few methods in the H3.6 and AMASS datasets are conducted.

**Audience:**

No

**Broader Impact Concerns:**

From a technical perspective, limitations are discussed. From a potential missuse, the discussion os fairly small.

**Claims And Evidence:**

Yes

**Requested Changes:**

The most important change to me is to clarify whether the paper is more oriented towards the human motion modeling application or towards the methodology.

If it is the first case, then the motivation of the proposed methodology regarding the human motion modeling application has to be clarified. Also, the downstream tasks should be moved from the appendix to the main paper, and the advantage of the proposed method w.r.t. to the state-of-the-art (in general, prediction quality, inference time, etc) should be extensively discussed.

If it is the second case, then the experiments cannot be limited to the human motion modeling task, as we need to be sure that the proposed methodology can be used on other domains (e.g. speech, ECG, etc). Also, the methodoogical contributions w.r.t. [a] (closest model in the literature) have to better explain, since right now it seems that it is an implementation matter. Finally, if the paper presents a methodological contribution, there should also be a comparison with dynamical VAEs [b], since they are the equivalent of VAE to process sequential data, and I believe MAP could also be applied to them.

As is, I do not believe the paper will interest many TMLR readers (and this is why I selected "No" in the audience question)/ But if the choice is clarified and the paper adjusted accordingly, it is possible it would interest several TRML readers.

**Strengths And Weaknesses:**

**Strengths**
 - The idea of a hierarchical graph-convolution variational autoencoder is interesting
 - Experiments are conducted in two large-scale datasets of motion modeling.

 **Weaknesses**
 - The reason why the study is limited to motion prediction is unclear, and the methodology is not specifically taylored to this application. So either the main contribution is the application, and one must compare with other methods, or the main contribution is the methodology, in which case the contributions must be very clearly explained.
 - Even if from an implementation perspective, there is some novelty, the methodological contribution is incremental, as I understand it as an implementation of the method in [a] with graph convolutional layers.
 - The notation is very confusing: it is very unclear how the temporality of the signal is dealt with, and if the network archtiecture can accomodate signals of different length.
 - There is no mention to the dynamica VAEs [b], that allow to process sequential data, which is highly related to the submitted manuscript.

[a] Very deep VAEs generalise autoregressive models and can ourperform the on images, R. Child, ICLR 2021.

[b] Dynamical Variational autoencoders; A comprehensive Review, L. Girin et al. FnT ML 2021.

---

> ### Author Response · Authors · 2024-03-24
> **Response, and details of submitted revised version related to this feedback.**
>
> We thank the reviewer for their time and feedback. You raise a good point in the lack of clarity as to whether the paper is a methodological contribution or an application. Indeed, it is primarily intended as an application. However, since much of the literature has focused on narrow discriminative objectives of prediction or classification that the paper somewhat presents as methodological. A comprehensive re-structuring of the paper has been undertaken in the revision with this point in mind. As suggested, the downstream tasks have been moved to the main paper, and for a more comprehensive comparison we have included the downstream tasks after MAP from HMVAE (we have also removed one of the downstream models as it made illustration more cumbersome and added nothing in the way of insight).
>
> The temporality is handled through conversion of timepoints into features based on their frequency. Discrete Cosine Transform was used for this. This technically enables the reception of signals of different length, by simply ignoring the lower frequencies, however, this was not examined in this work.
>
> In the revision we have included, in related work, Rempe, Davis, et al. "Humor: 3d human motion model for robust pose estimation." Proceedings of the IEEE/CVF international conference on computer vision. 2021, which propose a dynamic VAE, with the difference that the input to the encoder also receives the target timepoint—and the decoder receives the previous timepoint—with the motivation of encouraging the latent space to to carry the right information. This inclusion made a richer discussion of literature and clearer motivation for this work.

---

### Review · Reviewer_Y6mj · 2024-03-11

**Summary Of Contributions:**

Based on the pre-studied hierarchical VAE, this paper proposed a hierarchical graph-convolutional VAE-based network that was specifically designed to predict future poses from past occluded poses. Two main neural network computational blocks are introduced: the Graph Convolutional Layer (GCL) and the Graph Convolutional Block (GCB). The GCL takes a graph as input and outputs a new graph with potentially different numbers of nodes and features. The GCB, on the other hand, takes a graph as input and outputs a graph of the same dimensionality. Equations are provided to mathematically define the operations of the GCL and GCB, including the learnable parameters involved. The use of a Gaussian Error Linear Unit (GeLU) activation function and a learnable residual weighting ($\alpha$) for mitigating covariate shift is also mentioned. The experiments demonstrate the proficiency of the HG-VAE model in several aspects.

**Audience:**

Yes

**Claims And Evidence:**

Yes

**Requested Changes:**

See my comments above

**Strengths And Weaknesses:**

Strength:

The proposed approach combines hierarchical latent variables with a graph-convolutional structure for modeling human motion. The hierarchical structure of the model allows for the learning of motion patterns at multiple levels of abstraction, capturing both local and global dependencies in human motion. This enables the model to generate coherent and realistic motion sequences. By leveraging graph-convolutional operations, the model can effectively capture the spatial relationships between joints in the human body, which are crucial for modeling complex human motion. This architecture is well-suited for tasks that involve structured data like human pose sequences. The experiments demonstrate that the HG-VAE model outperforms baseline models, such as conventional VAEs and HM-VAE, in terms of mean squared error (MSE) and likelihood. This indicates that the proposed model can effectively model and reconstruct human motion.

Weakness:

The novelty of the proposed method seems limited. The core of the submission is their proposed network that is embedded in Hierarchical VAE, which has been well studied, with assumptions of Gaussians for all the conditional probability, but it seems to lack of motivation and details of the network development, such as why and how such networks can handle the application.

The proposed model's complexity, especially with the use of hierarchical latent variables and graph-convolutional structures, may result in high computational costs, limiting its scalability to larger datasets or real-time applications. The training time of 5000 epochs on a GPU suggests that the model may be computationally intensive. The training details mention the use of gradient clipping and a warm-up procedure for the KL divergence, indicating that the model may be sensitive to hyperparameters and require careful tuning.

---

> ### Author Response · Authors · 2024-03-24
> **Response, and details of submitted revised version.**
>
> Thank you to the reviewer for your time and feedback.
>
> Your comment on the motivation and development has been considered and appreciated, and a rewriting of the motivation has been undertaken. For your convenience, that is summarized here: although hierarchical VAEs have been well studied, there is no comparable literature that examines learnable contractions and expansions for graph networks, both spatially and temporally throughout different levels of the hierarchy. The closest comparison, HMVAE, simply pools joints precluding the learning of non-linear spatial relationships, and further does not examine the effect of depth (one important component of the enabling of depth is the rezero skip connections, which has been emphasised given your feedback).
>
> On computational complexity, you are indeed correct that a long training time is required. However, for a generative model of this capacity, both a very long train time, and instability in training is expected—which is mitigated by a ‘warm-up’. The good news is that this computational complexity (or instability) does not extend to inference. For instance, sampling, calculating posterior, and a step gradient ascent on the model’s posterior each only require a half-forward-pass through the model. This point has been expressed in the revision.

---

### Decision · Action_Editor_vuS5 · 2024-04-30

**Recommendation:** Reject

**Comment:**

The proposed paper presents a hierarchical graph convolutional VAE (HG-VAE) for modeling human motion. The HG-VAE is intended to be able to detect out-of-distribution inputs and handle missing data. Experiments and ablation studies are conducted on the H3.6 and AMASS datasets.

The reviewers agree that the paper in its current form is missing experimental comparisons to existing work and motivational reasoning for design choices in the context of the motion prediction application. As such, the current decision from the reviewers is to reject the paper. With a stronger experiments section and further reasoning for the architecture design, the reviewers believe that the paper would be of interest to the community.

**Audience:**

Although the paper may be of interest to some audience members in TMLR, the reviewers all agree that it would be made more impactful if significant comparisons to existing methods are added. One reviewer pointed out that "more general insights (e.g. another task with the same data or a different kind of data) would significantly enlarge the amount of people interested in reading this paper."

**Claims And Evidence:**

The reviewers are in agreement that the paper is lacking comparison experiments to existing literature to support the claims made in the manuscript. It is also not clear from the paper why certain design decisions work for the particular application of human motion modeling.